# All-optical control of long-lived nuclear spins in rare-earth doped nanoparticles

D. Serrano[1], J. Karlsson[1], A. Fossati[1], A. Ferrier[1,2] & P. Goldner[1]

Nanoscale systems that coherently couple to light and possess spins offer key capabilities for quantum technologies. However, an outstanding challenge is to preserve properties, and especially optical and spin coherence lifetimes, at the nanoscale. Here, we report optically controlled nuclear spins with long coherence lifetimes ($T_2$) in rare-earth-doped nanoparticles. We detect spins echoes and measure a spin coherence lifetime of 2.9 ± 0.3 ms at 5 K under an external magnetic field of 9 mT, a $T_2$ value comparable to those obtained in bulk rare-earth crystals. Moreover, we achieve spin $T_2$ extension using all-optical spin dynamical decoupling and observe high fidelity between excitation and echo phases. Rare-earth-doped nano-particles are thus the only nano-material in which optically controlled spins with millisecond coherence lifetimes have been reported. These results open the way to providing quantum light-atom-spin interfaces with long storage time within hybrid architectures.

[1] Université PSL, Chimie ParisTech, CNRS, Institut de Recherche de Chimie Paris, 11, rue Pierre et Marie Curie, 75005 Paris, France. [2] Sorbonne Université, Campus Pierre et Marie Curie, 4 place Jussieu, 75005 Paris, France. Correspondence and requests for materials should be addressed to P.G. (email: philippe.goldner@chimieparistech.psl.eu)

Quantum systems with spin qubits that can be optically controlled allow efficient qubit initialization and readout, and quantum gate operations[1]. Moreover, photonic quantum states can be mapped to and/or entangled with spin qubits for storage and processing[2–4]. Such schemes are investigated in solid-state systems like colour centres in diamond, quantum dots in semi-conductors, and rare-earth-doped crystals. Targeted applications include quantum memories for light[2,5,6] or microwave photons[7], and quantum processors[1]. In this respect, crucial advances are expected at the nanoscale that include single qubit control and readout[8], multiple qubit gate operation[1,9], and extremely sensitive and localized sensing and imaging[10]. Strongly enhancing light–matter interactions using nano- or micro-cavities[11,12], or coupling different quantum systems to build hybrid devices with an optical interface[13,14] are other attractive possibilities. Optical control of spins can also be useful in nanoscale systems. Optical excitations are faster than direct radio-frequency (RF) excitations because they take advantage of strong optical transitions[15], while ensuring spatial selectivity because of light's much shorter wavelength. It may also lead to simpler fabrication of devices by avoiding incorporating antennas in proximity to the spins.

However, coherence lifetimes are often significantly shortened in nano-materials[16,17], impairing their use for quantum technologies. Indeed, surface effects, and/or high concentration of defects or impurities due to the synthesis process can cause strong dephasing mechanisms[16]. The latter can be partially cancelled in nanostructures embedded in bulk crystals[11,18]. For rare earths, this approach has led to promising demonstrations, including single spin coherent control[19] and quantum storage[20]. However, freestanding nanoparticles have a higher flexibility for integration with other systems. For example, nanodiamonds containing NV centers and rare-earth-doped nanoparticles have been integrated in high-finesse, fibre-based micro-cavities[21,22], to increase fluorescence rates through the Purcell effect. This enables fast single qubit state readout or efficient single photon emission. Other hybrid structures for quantum technologies have been proposed like nanodiamonds deposited on an active substrate[23] or interacting with plasmons in metallic particles[24]. Furthermore, nanoparticles could also enable photon and phonon density of states engineering to create bandgaps and achieve longer optical and spin population and coherence lifetimes[25–27].

In the following, we investigate the nuclear spin coherence of rare-earth dopants in nanoparticles at low temperatures. These materials have unique properties for nanoscale systems, showing narrow optical linewidths, down to 45 kHz at 1.3 K, and limited spectral diffusion[28]. This is favourable to coupling to high-finesse optical cavities and using electric dipole–dipole interactions for quantum gate implementation. In these nanoparticles, we now demonstrate nuclear spin coherence lifetimes from $1.3 \pm 0.2$ ms up to $8.1 \pm 0.6$ ms in $Eu^{3+}$-doped $Y_2O_3$ nanoparticles using a fully-optical protocol, which includes spin echo and spin

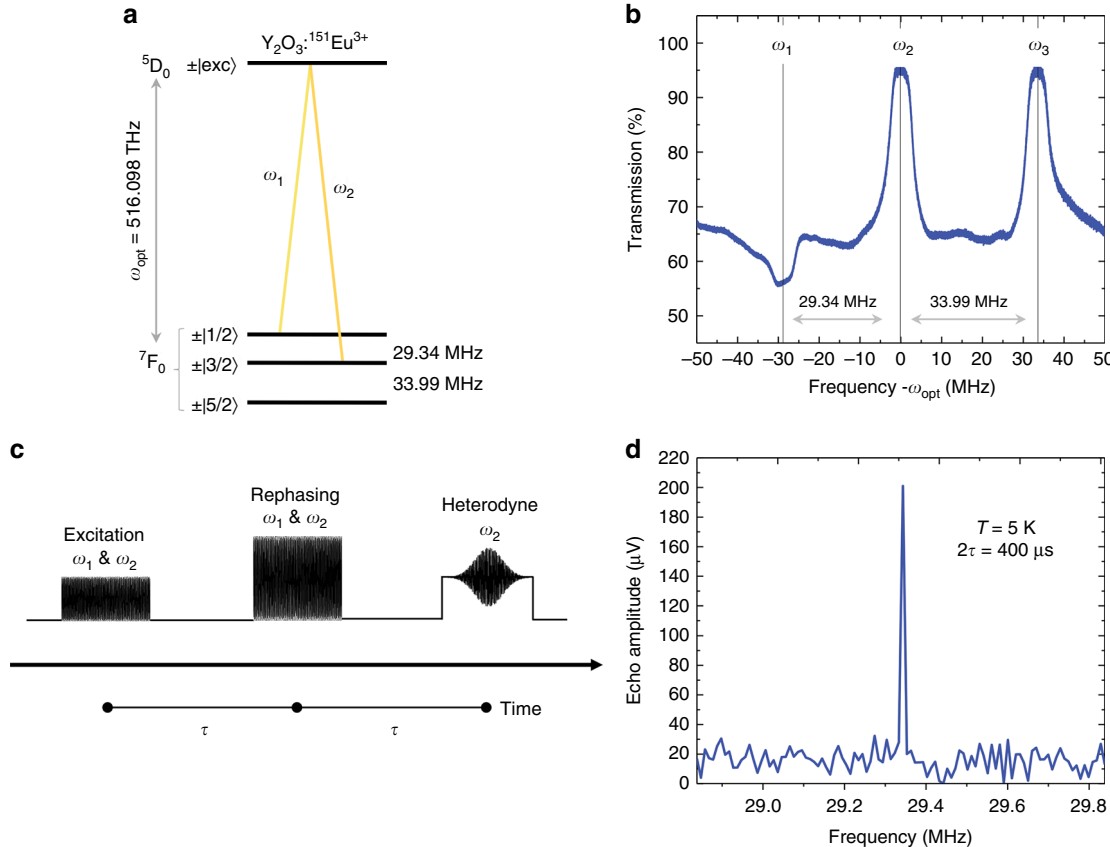

**Fig. 1** All-optical nuclear spin coherence investigation in $^{151}Eu^{3+}$-doped $Y_2O_3$ nanoparticles. **a** $^{151}Eu^{3+}$ ground-state hyperfine structure in $Y_2O_3$. Two-color laser pulses (at $\omega_1$ and $\omega_2$ frequencies) resonant with the $^{7}F_0 \rightarrow {}^{5}D_0$ transition at 580.883 nm create coherent states between the $\pm|1/2\rangle$ and $\pm|3/2\rangle$ nuclear spin levels. $\pm|exc\rangle$ represents the excited state hyperfine levels. **b** Optical transmission spectrum after optical pumping. Ground-state population initialization to $\pm|1/2\rangle$ corresponds to a lower transmission at $\omega_1$. High transmission (~95%) at 0 ($\omega_2$) and 33.99 MHz ($\omega_3$) evidences efficient population depletion in the $\pm|3/2\rangle$ and $\pm|5/2\rangle$ levels. $\omega_{opt} = 516.098$ THz (580.883 nm). **c** All-optical spin-echo sequence with heterodyne detection. Each sequence is preceded by optical pumping and followed by chirped pulses to reset the spin population to equilibrium. **d** Fast Fourier transform of the heterodyne signal revealing the spin echo at 29.34 MHz

dynamical decoupling (DD). High fidelity between excitation and echo phases is moreover observed, as required for quantum storage. These results suggest that rare-earth-doped nanoparticles, presenting both narrow optical and spin linewidths, could find multiple applications in optical quantum technologies.

## Results

**Spin coherence in rare-earth-doped nanoparticles**. Experiments were carried out on 0.5 % $Eu^{3+}:Y_2O_3$ nanoparticles of $400 \pm 80$ nm composed of $130 \pm 10$ nm crystallites obtained by homogeneous precipitation and high temperature annealing[27]. The particles were placed in a cryostat in the form of a powder and excited by laser pulses (see Methods). With a nuclear spin $I = 5/2$, the $^{151}Eu$ isotope presents three doubly degenerated ground-state nuclear spin levels at zero magnetic field (Fig. 1a). To probe the $\pm|1/2\rangle \leftrightarrow \pm|3/2\rangle$ hyperfine transition, the thermally distributed ground-state population was first initialized by optical pumping to the $\pm|1/2\rangle$ level for a subset of ions within the inhomogeneously broadened optical absorption line (Fig. 1b). Spin coherent states were subsequently created and rephased following an all-optical spin-echo sequence[29,30], using two-color pulses at frequencies $\omega_1$ and $\omega_2$ (Fig. 1c). A weak single-frequency pulse was applied at time $2\tau$ with frequency $\omega_2$ to convert the spin coherence into an optical coherence at $\omega_1$. This resulted in a beating at $\omega_2-\omega_1$ on the photodiode signal that was revealed with a signal to noise ratio (SNR) of about 10 by a fast Fourier transform (FFT) as displayed in Fig. 1d.

The spin-echo sequence was first used to determine the inhomogeneous broadening of the $\pm|1/2\rangle \leftrightarrow \pm|3/2\rangle$ transition, which was found equal to $107 \pm 8$ kHz (Fig. 2a). This value, identical to that reported on $Y_2O_3:Eu^{3+}$ bulk crystals[31] and ceramics[32], reflects the high crystalline quality of the particles. The decay of the spin-echo amplitude as a function of the increasing pulse separation reveals a coherence lifetime of $1.3 \pm 0.2$ ms (Fig. 2b), corresponding to a homogeneous linewidth $\Gamma_h = (\pi T_2)^{-1}$ of 250 Hz. This spin coherence lifetime is one order of magnitude lower compared to $Eu^{3+}:Y_2O_3$ bulk transparent ceramics ($T_2 = 12$ ms[32]) and $Eu^{3+}:Y_2SiO_5$ bulk single crystals ($T_2 = 19$ ms[33]). In contrast, the nanoparticles' optical coherence lifetime is two orders of magnitude lower than the bulk values: $T_{2opt} = 7$ μs[28] versus $T_{2opt} = 1.1$ ms (C. W. Thiel, personal communication). Thus, the spin coherence is much more preserved when scaling down in size than the optical coherence. This is consistent with a previous analysis in which we proposed that optical dephasing is mainly due to perturbations related to surface electric charges[28]. These charges have, however, little influence on nuclear spin transitions as the ratio between optical and nuclear Stark coefficients is expected to be about 5 orders of magnitude[34]. This suggests that magnetic perturbations are responsible for the dephasing of the spin transition. Indeed, under a weak magnetic field, the homogeneous linewidth decreases and reaches 110 Hz at 9 mT (Fig. 2c). This variation can be modelled by magnetic dipole–dipole interactions between $Eu^{3+}$ spins and defects carrying electron spins (Fig. 2c and Supplementary Discussion). A small magnetic field reduces the dipole–dipole interaction Hamiltonian to secular terms, which in turn reduces $Eu^{3+}$ spin frequency shifts due to defect spin flips and, therefore, dephasing. Quantitative analysis was performed assuming that $Eu^{3+}$ spin dephasing is due to frequency shifts following a Gaussian distribution. The inferred defect concentration, $6.4 \times 10^{17}$ cm$^{-3}$ or 25 ppm relative to Y, also indicates that spin $T_2$ could be increased in higher quality samples[35].

**All-optical spin dynamical decoupling**. A well-known approach to control dephasing is DD[36]. Here, a train of $\pi$ pulses is applied

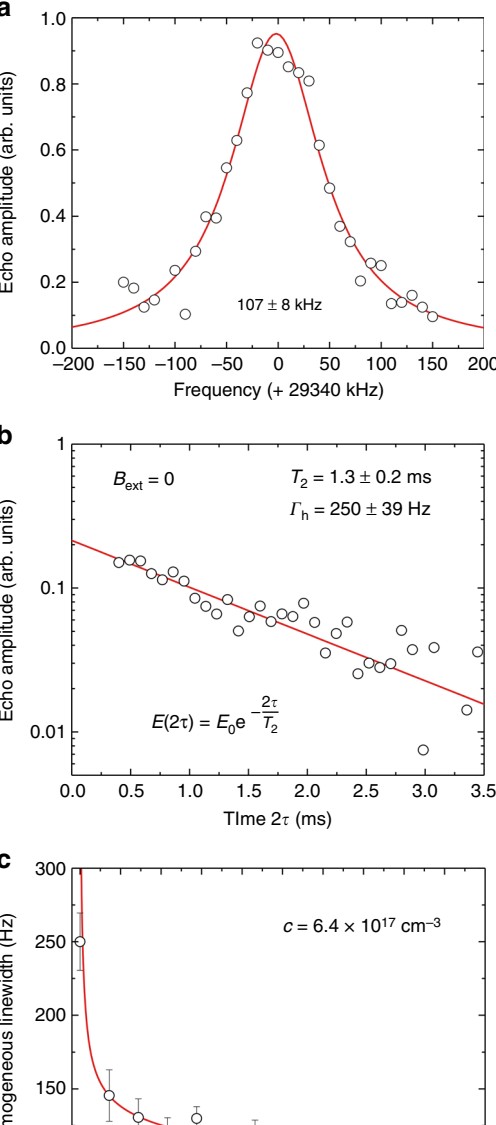

**Fig. 2** $^{151}Eu^{3+}$ spin inhomogeneous and homogeneous linewidths. **a** Inhomogeneous linewidth of the $\pm|1/2\rangle \leftrightarrow \pm|3/2\rangle$ spin transition obtained by monitoring the spin-echo amplitude as a function of the frequency detuning $\omega_2-\omega_1$ for a fixed time delay $2\tau$ of 400 μs (circles). Solid line: Lorentzian fit corresponding to a full width at half maximum of $107 \pm 8$ kHz. **b** Spin-echo decay at zero magnetic field. A single-exponential fit yields a coherence lifetime $T_2$ of $1.3 \pm 0.2$ ms, corresponding to a $\Gamma_h = 250$ Hz homogeneous linewidth. **c** Homogeneous linewidth evolution under an applied external magnetic field. A fast decrease in $\Gamma_h$ is observed for weak fields, corresponding to a coherence lifetime increasing from 1.3 ms to 2.9 ms (Supplementary Fig. 6). Solid line: modelling by interactions with defects carrying electron spins at a concentration of $6.4 \times 10^{17}$ cm$^{-3}$ (see Supplementary Discussion). Error bars and uncertainties correspond to ±1 standard error

to refocus frequency shifts due to fluctuations that are slow compared to the pulse separation. This principle was applied but with $\pi$ pulses corresponding to two-color laser pulses instead of the usual RF ones[33]. To the best of our knowledge this the first demonstration of an all-optical spin DD. A crucial point for DD,

is the phase coherence of the $\pi$ pulses. We achieved it by generating the two frequency shifted laser beams using a single acousto-optic modulator (AOM) and having them spatially overlap (see Methods). This ensured a highly stable relative phase between the two lasers beams and therefore phase coherent excitation, rephasing, and detection of the spins coherence.

The CPMG (Carl-Purcell-Meiboom-Gill)[37] DD sequence used in our experiments is shown in Fig. 3a. Coherence lifetimes extended by DD, $T_{2DD}$, were determined by recording the spin-echo amplitude vs. the total evolution time ($n \times \tau_{DD}$). This is efficient in preserving coherences along the $x$-axis of the Bloch sphere, but not those along the $y$-axis. This effect is due to the accumulation of errors in the $\pi$ pulse areas that have a larger effect for Y coherences than for X ones. In our powder, such pulse area errors are expected to be particularly high because of the random light scattering and orientation of the particles, which further increases the spread in spin Rabi frequencies. Indeed, significant

increase in coherence lifetime over the two-pulse echo value of 1.3 ms were achieved only for Y excitations (Fig. 3b). The $\pi$ pulse delay $\tau_{DD}$ was then varied, resulting in $T_{2DD} = 8.1 \pm 0.6$ ms for the optimal value $\tau_{DD} = 300$ μs, a 6-fold increase compared to the two-pulse echo $T_2$ (Fig. 3c). $T_{2DD}$ variation with $\tau_{DD}$, shown in Fig. 3d, can be explained by a balance between short $\tau_{DD}$ delays implying a higher number of pulses during a given evolution time and, therefore, accumulating pulse areas errors, and long delays that are less efficient in refocusing fluctuations[33] (see Supplementary Discussion). We also noted that applying a field of 0.7 mT decreased $T_{2DD}$, in opposition to $T_2$ (Fig. 3e). This could be explained by an increase in pulse errors when the transition broadens under magnetic field (Supplementary Fig. 2).

We finally investigated the variation of spin-echo phase as a function of the initial excitation phase in the 2 pulse and DD all-optical sequences (see Methods). They were found to be highly correlated, even for the DD case, in which a lower SNR was

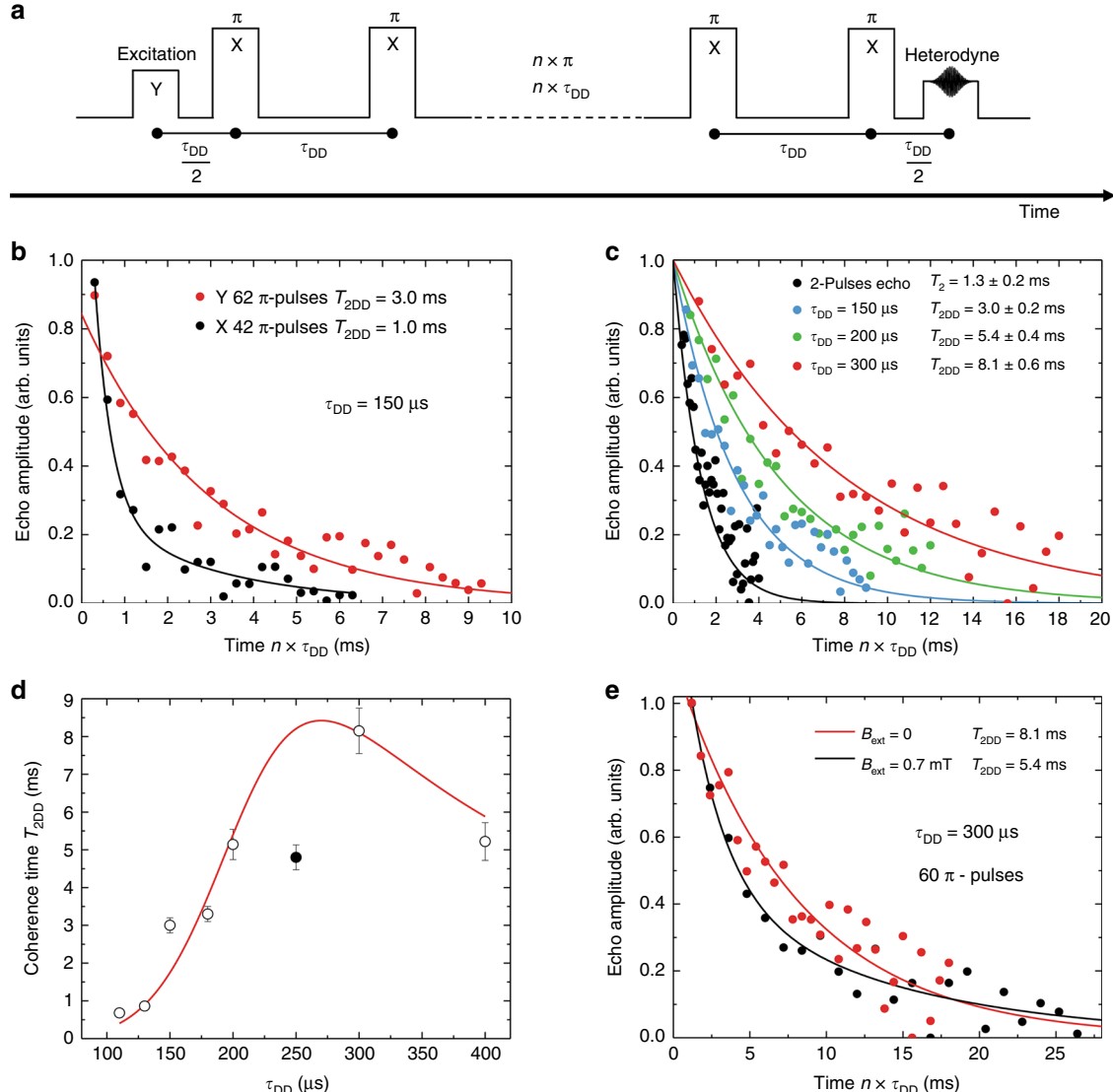

**Fig. 3** All-optical dynamical decoupling. **a** CPMG sequence with optical 2-color excitation and rephasing $\pi$ pulses. The initial excitation pulse has a Y phase and the $\pi$ pulses an X phase. This is obtained by varying the relative phase between the two frequency components of the optical pulses. **b** Echo decays (circles) for different initial phases and exponential fits (lines). A much lower $T_{2DD}$ is observed for an X initial phase (~1.0 ms) than for an Y one (~3.0 ms). This is due to the accumulation of pulse errors and confirms that our DD sequence behaves as a CPMG one. **c** Spin-echo decays (circles) obtained for $\tau_{DD}$ = 150, 200 and 300 μs with $n \leq 60$, and exponential fits (lines). **d** Experimental (circles) and modelled (line) $T_{2DD}$ evolution as a function of $\tau_{DD}$. (see Supplementary Fig. 5). The data point represented by the black circle was discarded for the fit. **e** Spin-echo decays (circles) with and without a weak magnetic field. Solid line: exponential fits. Error bars and uncertainties correspond to ±1 standard error

achieved (Fig. 4, Supplementary Fig. 7 and Supplementary Fig. 8). This confirmed the fully coherent character of the spins driving and detection. These experiments can also be considered as an optical memory with spin storage, with the initial and final light fields at $\omega_1$ being input and output signals (Figs. 1c and 3a). The high correlations of Fig. 4 then correspond to a high phase fidelity, an essential requirement towards an optical memory operating at the quantum level. In this respect, further investigations on the noise level introduced by all-optical DD will be necessary to assert the possibility of long time high-fidelity storage with spins. It will also be important to achieve faithful all-optical operations on spin states for quantum memories and processors. Suitable schemes using resonant two-color excitations have been proposed for rare-earth-doped crystals[38,39], reaching experimental $\pi$ pulse fidelity of 96%[40]. Similar results could be achieved in a single $Eu^{3+}$-doped nanoparticle, where interactions between light and ions are well defined, as long as optical pulses much shorter than the optical $T_2$ (7 μs[28]), but still longer than the inverse of the hyperfine splitting $((29 \times 10^6)^{-1} = 34$ ns) are used. The corresponding high Rabi frequencies could be obtained by placing the particle in an optical micro-cavity[21].

While the spin coherence lifetimes reported here are within a factor of ten from bulk values, they could still be increased in samples with lower content of magnetic defects or by polarising them at lower temperatures and higher magnetic fields. Moreover, at the single particle level, $T_2$ could be further improved by several orders of magnitude by taking advantage of reduced

pulses area errors in DD and using clock transitions that appear in europium and other rare-earth ions under suitable magnetic fields[41]. This could open the way to nanoscale quantum light–matter-spin interfaces, useful for quantum memories with processing capabilities, hybrid opto-mechanical systems, or coupling to optical micro-cavities. Nanoparticles doped with essentially any rare-earth ion can also be synthesized in different size, shape and layered structures, as shown by their huge development as luminescent probes[42]. Although quantum grade materials are very demanding, our results suggest that rare-earth ion-doped nanoparticles could be an extremely versatile platform for nanoscale quantum technologies.

## Methods

**Nanoparticles synthesis and structural characterization.** 0.5% $Eu^{3+}$:$Y_2O_3$ nanoparticles with 400 ± 80 nm average diameter and 130 ± 10 nm crystallite size were grown by homogeneous precipitation[27]. An aqueous solution of Y $(NO_3)_3$•$6H_2O$ (99.9% pure, Alfa Aesar), $Eu(NO_3)_3$•$6H_2O$ (99.99% pure, Reacton), and urea $(CO(NH_2)_2 > 99\%$ pure, Sigma) was first heated at 85 °C for 24 h in a Teflon reactor, yielding $Eu^{3+}$:$Y(OH)CO_3$.n$H_2O$ particles in suspension. The metal and urea concentrations were 7.5 mmol $L^{-1}$ and 0.5 mol $L^{-1}$. After cooling to room temperature, the carbonate particles were collected via centrifugation, washed with distilled water once and absolute ethanol twice to remove the byproducts, and finally dried at 80 °C for 24 h. They were calcined at 1200 °C during 6 h (heating rate: 3 °C $min^{-1}$) to obtain $Eu^{3+}$:$Y_2O_3$ particles. The body-centered cubic $Y_2O_3$ structure (Ia-3 space group) of the particles and their average crystallite size were determined by X-ray diffraction. No evidence of other parasitic phases was found. The morphology, size, and dispersion of the particles were obtained by scanning electron microscopy (Supplementary Fig. 1).

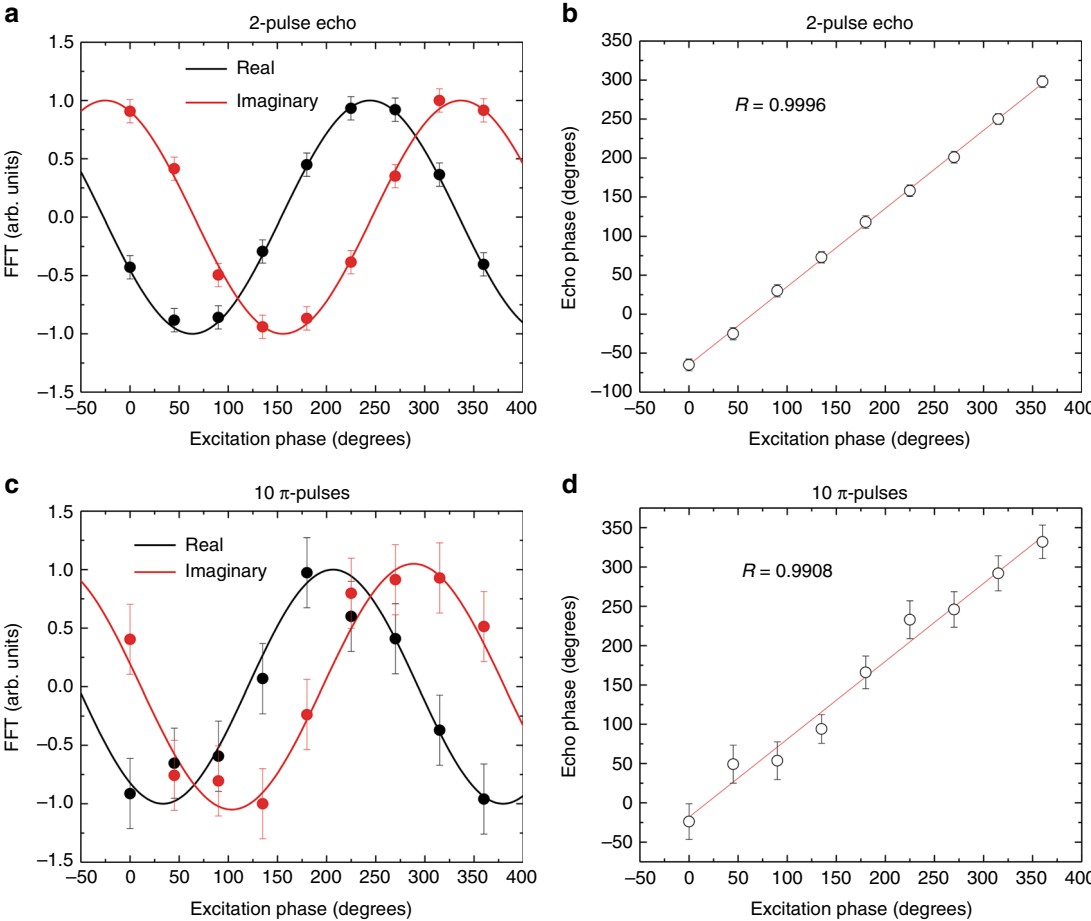

**Fig. 4** Echo phase correlation. **a**, **c** Real and imaginary parts of the spin-echo signal FFT (circles) as a function of the excitation pulse phase for a two-pulse echo and a DD sequence ($\tau = 300$ μs, $\tau_{DD} = 150$ μs, $n = 10$). Solid lines: fit with a sine function. Error bars were estimated from the signal to noise ratio in Supplementary Fig. 7 and Supplementary Fig. 8. **b**, **d** Echo pulse phase derived from FFT signals (**a**, **c**) as a function of the excitation pulse phase (see Methods). Lines: linear fit; $R$ correlation coefficient. Echo phase error bars were calculated by error propagation from the FFT signal to noise ratio

**Experimental setup**. The sample, consisting of an ensemble ($\approx 10^{11}$–$10^{12}$) of particles in form of loose powder, was placed between two glass plates with a copper spacer (~500 μm thickness) inside a He bath cryostat (Janis SVT-200). The excitation was provided by a Sirah Matisse DS laser, with a linewidth of ~150 kHz and operating at 516.098 THz (580.883 nm vac.) The laser beam was first sent through a double pass AOM with central frequency of 200 MHz (AA Optoelectronic MT200-B100A0, 5-VIS) followed by a single pass AOM (AA Optoelectronic MT110-B50A1-VIS) with a center frequency of 110 MHz. Both AOMs were driven by an arbitrary waveform generator (AWG) (Agilent N8242A) with 625 MS s$^{-1}$ sampling rate. The two-color pulses, generated by the single pass AOM, were coupled to a single-mode fiber in order to ensure spatial overlapping. The overlapped beams were then focused onto the sample, within the cryostat, with a 75 mm focal length lens and the scattered light collected with a 5 mm focal length lens mounted directly behind the sample holder. An avalanche photo diode (APD) (Thorlabs 110 A/M) was used as detector. A scheme of the experimental setup is displayed in Supplementary Fig. 3. The sample temperature was monitored with a temperature sensor (Lakeshore DT-670) attached to the sample holder with thermally conducting grease and tuned by controlling the helium gas flow and the pressure inside the cryostat. The cryostat was operated in gas mode to maintain a constant temperature of 5 K. Magnetic fields perpendicular to the laser beam propagation axis were applied by means of Helmholtz coils sitting outside the cryostat.

**Two-pulse spin-echo measurements**. Pulse areas in the two-pulse echo sequence were optimized to maximize the spin-echo signal. Data presented in this work were obtained with 100 μs-long pulses and total optical powers, $P_1 + P_2$, of the order of 120 mW, where $P_1$ and $P_2$ correspond to the optical powers applied to the $\omega_1$ and $\omega_2$ transitions, respectively. Although this input power is large compared to single crystal measurements, the scattering in the nanoparticles significantly reduces the input power incident in the sample. The power ratio between the two laser fields $P_1$ and $P_2$ was also optimized to maximize the spin-echo signal. Lower excitation power was used for the heterodyne pulse (~14 mW). Possible heating of the nanoparticles by the laser was checked by varying laser power and was found negligible in the measurements. The inhomogeneous linewidth of the 29 MHz spin transition was measured by monitoring the spin-echo signal as a function of the frequency difference ($\omega_2 - \omega_1$) in the two-color pulses for a fixed delay time $\tau$. The transition linewidth was estimated by a Lorentzian lineshape fit within an incertitude interval which was derived from the experimental SNR and the fit accuracy. The decay of the spin-echo signal with increasing $\tau$ was used to determine the nuclear spin coherence lifetime. The value was derived by single-exponential fit within an uncertainty also given by the experimental SNR and the fit accuracy.

**Dynamical decoupling and phase correlations measurements**. DD experiments were carried out with 20-μs-long $\pi$ pulses in order to access a large excitation bandwidth (about 50 kHz, half of the spin inhomogeneous linewidth) and short $\pi$-pulse separation times ($\tau_{DD}$). The preservation of the excitation phase coherence along the DD sequence was confirmed by the observation of stable beating patterns from a photodiode at the output of the fiber for times exceeding 30 s. The spin-echo phase was directly derived from the real (Re) and imaginary (Im) parts of the spin-echo signal FFT as

$$\theta_{echo} = \tan^{-1}\left(\frac{Im}{Re}\right) + n\pi \tag{1}$$

The error $\Delta\theta_{echo}$ was calculated by classical error propagation from the uncertainty associated to the real and imaginary FFT parts, $\Delta Re$ and $\Delta Im$. Those were estimated from the SNR in Supplementary Fig. 7 and Supplementary Fig. 8. As observed, the SNR is clearly larger in Supplementary Fig. 8 due to the weaker spin-echo signal obtained after 10 $\pi$-pulses, corresponding to a total evolution time of 1.5 ms compared to the total evolution time of 600 μs in the two-pulse echo case.

**Data Availability**. The data that support the findings of this study are available from the corresponding author upon reasonable request.

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

## Acknowledgements

This research work has received funding from the European Union's Horizon 2020 research and innovation programme under grant agreement No 712721 (NanOQTech). We thank John Bartholomew, Hugues de Riedmatten, and Thierry Chanelière for useful comments and discussions on the manuscript.

## Author contributions

J.K. developed the optical setup; D.S., J.K., A.Fo., and P.G. performed the experiments; D. S., J.K., A.Fe., and P.G. discussed, modelled, and analyzed the results; and D.S. and P.G. wrote the manuscript. All authors reviewed the manuscript.

## Additional information

**Competing interests:** The authors declare no competing interests.

