## [Peer Review File · Nature Communications]

Reviewers' comments:

Reviewer #1 (Remarks to the Author):

In this paper, the authors demonstrate all-optical control of the nuclear spins in Eu:Y2O3 nanoparticles and measure nuclear spin coherence times in the ms regime. While these coherence times are an order of magnitude shorter than what has been measured in the Eu doped bulk ceramics and single crystals, the excess dephasing is attributed to magnetic fluctuations due to defects carrying electron spins that could potentially be reduced by using higher quality samples. The authors then show that they can extend this coherence time by up to a factor of 6 using all-optical pulses to perform dynamical decoupling on the nuclear spins, which is the first demonstration of such a scheme. They also show that the spin echo phase is correlated with the phase of the excitation phase for both the two-pulse echo and with the dynamical decoupling sequence. The report of ms long coherence times and demonstration of all-optical dynamical in rare-earth doped nanoparticles is exciting and the authors put forth various ways in which these coherence times could be extended. However, I should point out that it is not that surprising that the nuclear spin coherence in nano-particles is high. In most nano-particles, including those made of diamond with NV centers, the electron and nuclear spin coherence is quite high, even though the optical coherence can be severely degraded. However, these results are promising for further pursuit of quantum light-matter interfaces incorporating REI doped nanoparticles.

I do have a few questions about the methods and results that the authors should address before making a decision for accepting this manuscript:

1. While the technique of all-optical control of spins is not new, it would be nice to see more analysis on the potential for using this (resonant stimulated raman transition) technique to perform high fidelity operations. I could be mistaken, but the citation given for the all-optical technique (23) does not seem to be appropriate here. In the cited paper, the authors use off-resonant raman pulses while in this work the authors are at both one and two-photon resonance. This reference could be replaced by a more relevant one (even citing previous demonstrations of all optical techniques by the authors of this work). While an off-resonant technique could be difficult in REIs due to the relatively weak transitions and inhomogeneous broadening, it seems that an on-resonant technique will lead to significant population in the excited state (especially in the case of imperfect pulses), which could negatively affect the fidelity of prepared spin states. Is this indeed a problem for this approach? Or is this problem mitigated in rare earths due to their long excited state lifetimes? Perhaps these questions have already been addressed in other publications..
2. Because the coherent raman readout is an inherently coherent process (i.e. you're looking for the signal beat at the difference frequency), won't the detected signal always have to be coherent? i.e. the coherent raman scattering will directly measure the coherence on the spin state? It then seems that you could still be preparing a very small coherence/mixed state on the spins, but are only reading out the coherent part of that state with this technique. In this case, it would seem that more investigations are needed before claiming that this is high fidelity storage on the spin state. Perhaps the claim being made in the paper is instead focused on preserving whatever coherence was excited on the spin state? While I agree that the detection must be "fully coherent", I would be hesitant to make claims that the preparation is fully coherent until it is further investigated. Similarly, I think the authors should make clear what they are claiming when they call this high fidelity storage. Alternatively, an explicit value/calculation of the fidelity would be useful.
3. How do you propose to mitigate the pulse areas associated with the optical pulses? Have you investigated the fidelity of a pulse theoretically?

4. On a similar note, how well can you create an effective $\pi/2$ pulse on the spins all-optically? Are the rabi frequencies used for the two arms of the lambda system equal? Assuming you're initialized into one spin state, I think the optimal $\pi/2$ pulse on these transitions should require a mixing angle of $\pi/8$ (i.e. a ratio between the arms of $\Omega_2/\Omega_1 = 1 + \sqrt{2}$). Is this correct or should the powers always be equal?

5. Does the experiment to measure spin inhomogeneous linewidth faithfully give the desired value? It seems that the pulse bandwidth and rabi frequency could have an effect on this measurement. Similarly, do you lose twice in echo amplitude when you detune the two-photon frequency? i.e. as you detune you will get a lower excitation on the spins ($\pi/2$ pulse) and then also a lower rephasing (π pulse)? In this case, would this method give a narrower inhomogeneous linewidth than the actual linewidth?

6. Aside from simplifying the experimental setup by removing the need for an RF antenna to drive the nuclear spins, one of the selling points of an all-optical scheme is the potential to perform fast spin operations by using strong optical transitions. In this work, the all-optical pulses first presented to drive the spins are ~ 100 us long, which I wouldn't consider especially short compared to the coherence time of ~ 1 ms. While the authors move to shorter pulses (20 us) for the DD protocol, they are still only exciting half of the spin inhomogeneous line which it seems would increase the error due to the DD pulses. Is the limit on pulse length in the presented experiment essentially the available laser power and the large amount of scattering from the nanoparticles?

7. The length of the pulses used is much longer than the coherence time of the optical transition. Does this have a negative effect on the achievable fidelity of effective spin pulses?

8. Does laser stability have a negative effect on measurable coherence times using this technique?

9. The authors mention that the transition broadens under magnetic field. What level of inhomogeneous broadening is observed?

10. Is 9mT the maximum field available experimentally? Do you have any predictions for the spin coherence at high fields (100mT and up) or lower temperatures that might help freeze out contributions from the electron spin bath?

11. The authors make claims that the magnetic field "strongly" or "drastically" decreases the spin linewidth, which seems to be an overly strong statement when it only decreases by a factor of two.

2. There are a few small typos throughout the manuscript and it could benefit from a careful proofreading.

A few examples:

1) rephasing is misspelled in Fig. 1c.

2) the temperature in Fig. 1d is listed as 6K, but given as 5K in the paper. Is this correct?

3) in the caption for figure 3, rephasing has been (most likely) autocorrected to rephrasing.

4) In methods section: "cryostat was operated in gaz mode" should be "cryostat was operated in gas

Reviewer #2 (Remarks to the Author):

Dear authors,

The manuscript by Serrano et al describes measurements on nano-particles, more precisely Europium (a rare-earth) crystals, that show that the spin coherence time is only weakly degraded with respect to that of larger crystals. This result is relevant in view of recent interest in impurity-doped nanocrystals, in particular for quantum technology based on single qubit control and readout. The manuscript is interesting, but I am not (yet) convinced that it has sufficient appeal to warrant publication in a high-profile journal such as Nature Communications.

First, I do not think that the abstract makes a good case for why nanocrystals (as opposed to standard bulk crystals) are required to advance certain quantum technologies based on individual impurities. I would have imagined to read more about, say, nano/micro cavities (and the importance of Purcell-enhanced emission) in the Faraon group, or the idea of increasing lifetimes and coherence times due to the modification of the phonon density of states (mentioned in a recent review by Becher for SiV in diamond, or pursued by the Tittel group with rare-earth-doped crystals). Some additional points are made in the introduction, but I suggest expanding both parts to make a better case.

Furthermore, I am also a bit disappointed that the spin lifetime is a factor of 10 shorter than those observed in corresponding bulk crystals. Page 3 mentions a reduction by one order of magnitude, which I find quite significant even though the authors present it as a remarkable success. In case it is indeed remarkable, e.g. much better than what has been reported previously, then this should be supported by appropriate language and references. I also note that the abstract mentions comparable coherence times for nano and bulk crystals, which is not supported by the data.

From a technical point of view, the investigation is well performed and the manuscript well written. I particularly like the discussion of spin versus optical T2 times, and the analysis of transition sensitivity to magnetic and electric perturbations. A few minor points that the authors should consider are as follows:

It may be useful in the last paragraph on page 2 (when the Eu:Y2O3 nanoparticles are introduced) to refer to the synthesis in the methods section. Also, please include information about the purity of the precursors.

In addition, please discuss how convinced you are that the measured temperature is indeed the local temperature of the nano crystals. I imagine that it is very easy to cause heating during optical excitation (in particular given the large excitation powers used), which could explain the reduced spin T2 time.

Reviewer #3 (Remarks to the Author):

This paper demonstrates millisecond long nuclear spin coherence times in earth doped nanoparticles, which is a record for nuclear spins in nanoparticles, and close to the bulk values. Moreover, while all-optical spin control is not new, the authors show that all-optical dynamical decoupling can be applied and leads to an increased coherence time, which is the first time to my knowledge. Therefore, I recommend publication in Nature Comms.

I just have a few questions and minor suggestions:

1. It is not clear how many nanoparticles are being measured. Please give an estimate. How many Eu ions in total does this correspond to?

2. It would also be useful to quote the average number of Eu ions per particle, and the number of magnetic defects per particle inferred from the measurements.
3. Have the authors tried to use other dynamical decoupling sequences, such as XY8? Since some sequences are more robust against pulse errors than CPMG, further increase in coherence times could be achieved.

Reviewer #1 (Remarks to the Author):

In this paper, the authors demonstrate all-optical control of the nuclear spins in Eu:Y2O3 nanoparticles and measure nuclear spin coherence times in the ms regime. While these coherence times are an order of magnitude shorter than what has been measured in the Eu doped bulk ceramics and single crystals, the excess dephasing is attributed to magnetic fluctuations due to defects carrying electron spins that could potentially be reduced by using higher quality samples. The authors then show that they can extend this coherence time by up to a factor of 6 using all-optical pulses to perform dynamical decoupling on the nuclear spins, which is the first demonstration of such a scheme. They also show that the spin echo phase is correlated with the phase of the excitation phase for both the two-pulse echo and with the dynamical decoupling sequence. The report of ms long coherence times and demonstration of all-optical dynamical in rare-earth doped nanoparticles is exciting and the authors put forth various ways in which these coherence times could be extended. However, I should point out that it is not that surprising that the nuclear spin coherence in nano-particles is high. In most nano-particles, including those made of diamond with NV centers, the electron and nuclear spin coherence is quite high, even though the optical coherence can be severely degraded. However, these results are promising for further pursuit of quantum light-matter interfaces incorporating REI doped nanoparticles.

We thank the reviewer for his/her very positive and encouraging comments.

We would like to point out that predicting spin (or optical) coherence lifetimes in nanomaterials is, in our opinion, challenging because nano-systems are prone to defects, including impurities. These defects can vary a lot between very different hosts, like rare earth doped crystals and diamond, which are moreover synthesized in completely different ways. The centers are quite different too. For example, excited state T1 dominates NV optical linewidths and can be as short as a few ns in nanodiamonds; rare earth optical linewidths have only a negligible (300 Hz) contribution from T1 and are determined by interactions with phonons, defects, etc.. Optical and spin variations can therefore be due to quite different processes. We also note that NV centers in nanodiamond exhibit electron spin T₂ in the few μs range (without DD) [H. S. Knowles et al., Nat. Mater. (2013), Ref. 8] whereas, ≈2 ms have been found in the best bulk samples [G. Balasubramanian et al., Nat. Mater. (2009)]. Finally, we report here for the first time spin coherence lifetime in a rare earth nanostructure.

I do have a few questions about the methods and results that the authors should address before making a decision for accepting this manuscript:

1. While the technique of all-optical control of spins is not new, it would be nice to see more analysis on the potential for using this (resonant stimulated Raman transition) technique to perform high fidelity operations. I could be mistaken, but the citation given for the all-optical technique (23) does not seem to be appropriate here. In the cited paper, the authors use off-resonant Raman pulses while in this work the authors are at both one and two-photon resonance. This reference could be replaced by a more relevant one (even citing previous demonstrations of all optical techniques by the authors of this work).

We agree that references dealing with resonant excitation is more relevant to our work and we replaced the previous reference with

30. Ham, B. S., Shahriar, M. S., Kim, M. K. & Hemmer, P. R. Spin coherence excitation and rephasing with optically shelved atoms. *Phys. Rev. B* **58**, 11825–11828 (1998).

and

31. Guillot-Noël, O. *et al.* Hyperfine structure and hyperfine coherent properties of praseodymium in single-crystalline $\text{La}_2(\text{WO}_4)_3$ by hole-burning and photon-echo techniques. *Phys. Rev. B* **79**, 155119 (2009).

While an off-resonant technique could be difficult in REIs due to the relatively weak transitions and inhomogeneous broadening, it seems that an on-resonant technique will lead to significant population in the excited state (especially in the case of imperfect pulses), which could negatively affect the fidelity of prepared spin states. Is this indeed a problem for this approach? Or is this problem mitigated in rare earths due to their long excited state lifetimes? Perhaps these questions have already been addressed in other publications.

We thank the reviewer for bringing to attention this question. High fidelity all-optical preparation of rare earth spin states has been investigated theoretically and experimentally in the context of quantum computing [Refs 1-3 below].

This is indeed an important point for the applications we envision for rare earth doped nanoparticles and we agree it should be mentioned in the manuscript. A detailed study of how accurately we can prepare spin states is however experimentally out of reach in a powder in which light-ion interactions are not well defined. In [2], it is concluded that in a sub-ensemble of ions prepared in a single spin state by spectral tailoring and with low inhomogeneous broadening, average fidelities up to 96% can be obtained for single qubit gates using resonant two-color optical excitations. Using single ions, even higher fidelities could be obtained [3]. In a single nanoparticle, these schemes could be directly applied assuming that lower signals could be compensated by using a micro-cavity, as proposed in [4]. In our particles, the excited state T_2 is shorter than those considered in Refs. [1-3]. We ran Bloch simulations to quantify the impact of the reduced lifetime and found a fidelity of 0.9894 for a $\pi/2$ pulse of 250 ns ($\Omega = 2\pi \times 1.5$ MHz).

We added a discussion of fidelity on p. 6-7 and the references below (22, 39, 40, 41).

[1] I. Roos and K. Mølmer, "Quantum computing with an inhomogeneously broadened ensemble of ions: Suppression of errors from detuning variations by specially adapted pulses and coherent population trapping," 69, 022321 (2004).

[2] L. Rippe, B. Julsgaard, A. Walther, Y. Ying, and S. Kröll, "Experimental quantum-state tomography of a solid-state qubit," *Phys. Rev. A* **77**, 022307 (2008).

[3] A. Walther, L. Rippe, Y. Yan, J. Karlsson, D. Serrano, A. N. Nilsson, S. Bengtsson, and S. Kröll, "High-fidelity readout scheme for rare-earth solid-state quantum computing," *Phys. Rev. A* **92**, 022319 (2015).

[4] B. Casabone, J. Benedikter, T. Hümmer, F. Beck, K. de O. Lima, T. W. Hänsch, A. Ferrier, P. Goldner, H. de Riedmatten, and D. Hunger, "Cavity-enhanced spectroscopy of a few-ion ensemble in $\text{Eu}^{3+}:\text{Y}_2\text{O}_3$," [arXiv:1802.06709](https://arxiv.org/abs/1802.06709) (2018).

2. Because the coherent Raman readout is an inherently coherent process (i.e. you're looking for the signal beat at the difference frequency), won't the detected signal always have to be coherent? i.e. the coherent Raman scattering will directly measure the coherence on the spin state? It then seems that you could still be preparing a very small coherence/mixed state on the spins, but are only reading out the coherent part of that state with this technique. In this case, it would seem that more investigations are needed before claiming that this is high fidelity storage on the spin state. Perhaps the claim being made in the paper is instead focused on preserving whatever coherence was excited on the spin state? While I agree that the detection must be "fully coherent", I would be hesitant to make claims that the preparation is fully coherent until it is further investigated. Similarly, I think the authors should make clear what they are claiming when they call this high fidelity storage. Alternatively, an explicit value/calculation of the fidelity would be useful.

The high fidelity we refer to concerns the initial and final light fields. Although the echo (or DD) process is certainly coherent, the echo phase can have no fixed relation with the phase of the initial coherence from shot to shot. This is usually the case for optical echoes, because of the high laser phase noise. In our experiments, we prepare an initial two-color pulse with a given relative optical phase and after a few ms detect the phase of a heterodyne signal. This is repeated over many shots and the heterodyne signals averaged. We find a very stable relation between input and output relative optical phases for input phases between 0 and 360° . The whole process has therefore a high fidelity (which can be seen as a comparison between the output phases over many shots), which does not mean, as pointed out by the referee, that the storage in the spin itself is faithful, although some coherence must be stored in the spins. We clarified this in the last paragraph on page 6.

3. How do you propose to mitigate the pulse areas associated with the optical pulses? Have you investigated the fidelity of a pulse theoretically?

Pulse area errors could be very well mitigated by using composite or complex hyperbolic secant pulses as proposed and demonstrated for rare earth ions [1-2]. It is not clear however whether they could be efficient in our highly inhomogeneous system, and we did not try these pulses experimentally.

[1] I. Roos and K. Mølmer, "Quantum computing with an inhomogeneously broadened ensemble of ions: Suppression of errors from detuning variations by specially adapted pulses and coherent population trapping," *Phys. Rev. Lett.* **69**, 022321 (2004).

[2] L. Rippe, B. Julsgaard, A. Walther, Y. Ying, and S. Kröll, "Experimental quantum-state tomography of a solid-state qubit," *Phys. Rev. A* 77, 022307 (2008).

4. On a similar note, how well can you create an effective $\pi/2$ pulse on the spins all-optically? Are the Rabi frequencies used for the two arms of the lambda system equal? Assuming you're initialized into one spin state, I think the optimal $\pi/2$ pulse on these transitions should require a mixing angle of $\pi/8$ (i.e. a ratio between the arms of $\Omega_2/\Omega_1 = 1 + \sqrt{2}$). Is this correct or should the powers always be equal?

Yes, this is correct, and we experimentally optimized the power ratio between the two optical fields in order to achieve maximum echo amplitude. It was not possible to relate that to Rabi frequencies because the relative strengths of the optical transitions between ground and excited states are not known in $\text{Eu:Y}_2\text{O}_3$. To determine them, extensive hole burning experiments are required, which were out of the scope of the present study.

We added to the methods section (two-pulse echo measurements, p. 9-10) that the ratio between the two laser fields was optimized.

5. Does the experiment to measure spin inhomogeneous linewidth faithfully give the desired value? It seems that the pulse bandwidth and Rabi frequency could have an effect on this measurement. Similarly, do you lose twice in echo amplitude when you detune the two-photon frequency? i.e. as you detune you will get a lower excitation on the spins ($\pi/2$ pulse) and then also a lower rephasing (π pulse)? In this case, would this method give a narrower inhomogeneous linewidth than the actual linewidth?

The technique we use here to estimate the inhomogeneous linewidth, based on the spin echo amplitude evolution as a function of the two-color detuning, is equivalent to the field-swept echo technique used in pulsed electron spin resonance [1]. It is commonly used to determine inhomogeneously broadened electron spin lines in solids as a simple and highly sensitive alternative to direct absorption measurements. As our pulse bandwidth and estimated Rabi frequency are below 10 kHz, we do not expect them to influence the measurement above the experimental error bars. Since we are dealing with a transition with low inhomogeneous broadening, we assume that the pulse areas are constant over the linewidth.

[1] M. J. Colaneri et al., Enhanced Resolution of EPR Single-Crystal Spectral Parameters Using Field-Swept Electron Spin-Echo Spectroscopy, *Journal of Magnetic Resonance Journal of Magnetic Resonance, Series A* 102, 360 (1993).

6. Aside from simplifying the experimental setup by removing the need for an RF antenna to drive the nuclear spins, one of the selling points of an all-optical scheme is the potential to perform fast spin operations by using strong optical transitions. In this work, the all-optical pulses first presented to drive the spins are ~ 100 us long, which I wouldn't consider especially short compared to the coherence time of ~ 1 ms. While the authors move to shorter pulses (20 us) for the DD protocol, they are still only exciting half of the spin inhomogeneous line which it seems would increase the error

due to the DD pulses. Is the limit on pulse length in the presented experiment essentially the available laser power and the large amount of scattering from the nanoparticles?

In our experiments, we were indeed limited by the available laser power and the strong scattering in the powder. In principle, much shorter pulses could be used, limited by the hyperfine splitting (29 MHz, corresponding to about 34 ns). As already mentioned, this would be possible using a single particle. If it is located in a micro-cavity, very high Rabi frequencies could be achieved even with sub μ s pulses.

We added a sentence about this point in the discussion about fidelity on p. 6-7.

7. The length of the pulses used is much longer than the coherence time of the optical transition. Does this have a negative effect on the achievable fidelity of effective spin pulses?

Bloch simulations indeed show that this is the case. To reach very high fidelity, sub- μ s pulses could be used as discussed in Ref. 39 and 40, but this regime could not be reached in our experiments, as explained above.

We added a sentence about this point in the discussion about fidelity on p. 6-7.

8. Does laser stability have a negative effect on measurable coherence times using this technique?

Laser stability can be a limit for long evolution times since frequency drifts will prevent addressing the same ions during the sequence. In the present measurements, we are still far from seeing such effects, as shown by all-optical DD in bulk samples where T_2 could reach 45 ms (unpublished). In the future, we are considering improving our commercially stabilized laser by using a better reference cavity.

9. The authors mention that the transition broadens under magnetic field. What level of inhomogeneous broadening is observed?

A broadening of the order of 44 kHz/mT is observed for fields < 10 mT.

The figure below and a comment have been added to the SI on page 8.

10. Is 9mT the maximum field available experimentally? Do you have any predictions for the spin coherence at high fields (100mT and up) or lower temperatures that might help freeze out contributions from the electron spin bath?

Higher magnetic fields, up to 3 T, are available in our laboratory, but the broadening of the spin transition led to a decrease of the echo amplitude. Above 9 mT we could not record signals with sufficient signal to noise ratio. Lower temperatures would be indeed very interesting to study. Under a field of 100 mT and at 20 mK, the electron spins should freeze, and the main perturbation be due to ^{89}Y nuclear spin flips. Given their low gyromagnetic factor, they will not strongly polarize, even at 20 mK. We should therefore reach the value of the bulk samples, i.e. about 10 ms. This contribution could then be suppressed by DD and/or ZEF0Z transitions, as mentioned in the paper's conclusion. Hours of nuclear spin T_2 can be potentially reached according to Ref. 42. We added a comment on this on page 8.

11. The authors make claims that the magnetic field “strongly” or “drastically” decreases the spin linewidth, which seems to be an overly strong statement when it only decreases by a factor of two.

The terms strongly and drastically were not really chosen to mean that T_2 decreases a lot in absolute value, since, as well pointed out, it is a factor 2, but to underline that T_2 is sensitive to very small magnetic fields (as small as ~ 1 mT or 10 G). We attribute this effect to field dependent interactions with electron spins, as described in the manuscript.

For the sake of clarity, we now use 'fast' in Figure 2 caption .

12. There are a few small typos throughout the manuscript and it could benefit from a careful proofreading.

A few examples:

- 1) rephasing is misspelled in Fig. 1c.
- 2) the temperature in Fig. 1d is listed as 6K, but given as 5K in the paper. Is this correct?
- 3) in the caption for figure 3, rephasing has been (most likely) autocorrected to rephrasing.
- 4) In methods section: “cryostat was operated in gaz mode” should be “cryostat was operated in gas

We thank the reviewer for careful reading and apologize for the typos. They have been corrected and the manuscript has been carefully read.

Reviewer #2 (Remarks to the Author):

Dear authors,

The manuscript by Serrano et al describes measurements on nano-particles, more precisely Europium (a rare-earth) crystals, that show that the spin coherence time is only weakly degraded with respect to that of larger crystals. This result is relevant in view of recent interest in impurity-doped nanocrystals, in particular for quantum technology based on single qubit control and readout. The manuscript is interesting, but I am not (yet) convinced that it has sufficient appeal to warrant publication in a high-profile journal such as Nature Communications.

We thank the reviewer for his/her interest in our work. In the following, we answer the points raised to our best, including significant changes in the manuscript, and hope that the revised version will be suitable for publication.

First, I do not think that the abstract makes a good case for why nanocrystals (as opposed to standard bulk crystals) are required to advance certain quantum technologies based on individual impurities. I would have imagined to read more about, say, nano/micro cavities (and the importance of Purcell-enhanced emission) in the Faraon group, or the idea of increasing lifetimes and coherence times due to the modification of the phonon density of states (mentioned in a recent review by Becher for SiV in diamond, or pursued by the Tittel group with rare-earth-doped crystals). Some additional points are made in the introduction, but I suggest expanding both parts to make a better case.

We should indeed have mentioned these points. Following the reviewer’s suggestions, we have expanded on the advantages of nanoparticles for quantum technologies in the introduction on page 2. We now mention in more detail in more details coupling to optical cavities (note that Refs. 17 and 18 were already given in the original manuscript) and give another example of hybrid structures. Phonon and photon density of state engineering in nanoparticles has also been added.

The new references are:

20. Siyushev, P. *et al.* Coherent properties of single rare-earth spin qubits. *Nat. Commun.* **5**, 3895 (2014).
21. Zhong, T. *et al.* Nanophotonic rare-earth quantum memory with optically controlled retrieval. *Science* **357**, 1392–9 (2017).
22. Casabone, B. *et al.* Cavity-enhanced spectroscopy of a few-ion ensemble in Eu³⁺:Y₂O₃. (2018).
23. Johnson, S. *et al.* Tunable cavity coupling of the zero phonon line of a nitrogen-vacancy defect in diamond. *New J. Phys.* **17**, 122003 (2015).
25. Gong, J., Steinsultz, N. & Ouyang, M. Nanodiamond-based nanostructures for coupling nitrogen-vacancy centres to metal nanoparticles and semiconductor quantum dots. *Nat. Commun.* **7**, 1–10 (2016).
26. Lutz, T. *et al.* Modification of phonon processes in nanostructured rare-earth-ion-doped crystals. *Phys. Rev. A* **94**, 013801 (2016).
27. Jahnke, K. D. *et al.* Electron–phonon processes of the silicon-vacancy centre in diamond. *New J. Phys.* **17**, 043011 (2015).

Furthermore, I am also abit disappointed that the spin lifetime is a factor of 10 shorter than those observed in corresponding bulk crystals. Page 3 mentions a reduction by one order of magnitude, which I find quite significant even though the authors present it as a remarkable success. In case it is indeed remarkable, e.g. much better than what has been reported previously, then this should be supported by appropriate language and references. I also note that the abstract mentions comparable coherence times for nano and bulk crystals, which is not supported by the data.

As noted in the answer to the first reviewer, predicting spin coherence lifetimes in nanomaterials is, in our opinion, not straightforward. First, this has never been measured in a rare earth doped nanomaterial. Second, comparison with other nanoparticles, like nano-diamonds (ND), is also difficult: they are synthesized in completely different ways, hosting different defects and impurities. Still, electron spins in ND have T_2 of a few μs (without dynamical decoupling) as quoted in [H. S. Knowles *et al.*, *Nat. Mater.* (2013), Ref. 8], whereas 100s of μs , up to ms, are reported in high quality bulk samples [G. Balasubramanian *et al.*, *Nat. Mater.* (2009)]. This is a clear indication, given the decades of work on diamond, that long T_2 at the nanoscale is very challenging.

In this respect, we believe that observing nuclear spin T_2 between 1.2 and 2.9 ms (without DD), admittedly still significantly shorter than bulk values, is a key result that makes this system truly unique. As mentioned in the introduction, to our knowledge, it is the only nanomaterial showing spins with ms T_2 that can be addressed through narrow optical transitions (< 50 kHz). This opens the way to new nanoscale quantum light-matter interfaces and processors that are currently unavailable with other systems.

On page 3, we agree with the reviewer that the sentence beginning with ‘remarkably’ may be confusing. We refer here to the large difference between optical and spin T_2 , an interesting observation that supports our recent and new hypothesis of electric field perturbations as the

main dephasing process for optical transitions in this system (Ref. 29). However, as this is not the main point of the manuscript, we removed 'remarkably' on page 3.

We observe nuclear spin T_2 of 1.2 ms at zero field, 2.9 ms at 9 mT, and 8 ms with DD at zero field. The bulk value is 12 ms (zero field, no DD). We therefore think that 'comparable' is not misleading, although we agree that it has no clear quantitative meaning. We also note that high profile articles like "Observing bulk diamond spin coherence in high-purity nanodiamonds" by Knowles et al., Nature Materials 2013 (Ref. 8 in manuscript) reports on T_2 of about 6 μ s and 60 μ s with DD, whereas the very best bulk diamonds reach ms T_2 . The two other reviewers seemed to accept 'comparable' to describe our results, and we have replaced comparable by 'within a factor of 10' in the conclusion but kept it in the abstract.

From a technical point of view, the investigation is well performed and the manuscript well written. I particularly like the discussion of spin versus optical T_2 times, and the analysis of transition sensitivity to magnetic and electric perturbations.

We thank the reviewer for his/her positive comment.

A few minor points that the authors should consider are as follows:

It may be useful in the last paragraph on page 2 (when the Eu:Y2O3 nanoparticles are introduced) to refer to the synthesis in the methods section. Also, please include information about the purity of the precursors.

The detail of the synthesis and the purity of the precursors (3N for the Y precursor and 4N for the Eu³⁺ precursor) is given in Ref. 28. For completeness, we added this information to the methods section on page 9.

In addition, please discuss how convinced you are that the measured temperature is indeed the local temperature of the nano crystals. I imagine that it is very easy to cause heating during optical excitation (in particular given the large excitation powers used), which could explain the reduced spin T_2 time.

The powder is located in cold helium gas in a bath cryostat, which provides a high cooling power. The temperature quoted is that of a sensor attached to the sample holder. Heating by the laser is indeed possible when the preparation sequence (hole burning) has too many or too strong optical pulses, rather than the echo or DD pulses themselves. It is typically observed as a decrease in echo amplitude. We therefore kept the preparation sequence power low enough to avoid this problem and did not observe significant variations of T_2 between low and high echo or DD pulse intensities. We also studied in details temperature dependence of optical T_2 in the same nanoparticles (Ref. 29) that did not indicate significant deviation between sample and sensor temperatures. Since these measurements also use similar preparation sequences, we conclude that the temperature quoted in the manuscript is accurate.

Reviewer #3 (Remarks to the Author):

This paper demonstrates millisecond long nuclear spin coherence times in earth doped nanoparticles, which is a record for nuclear spins in nanoparticles, and close to the bulk values. Moreover, while all-optical spin control is not new, the authors show that all-optical dynamical decoupling can be applied and leads to an increased coherence time, which is the first time to my knowledge. Therefore, I recommend publication in Nature Comms.

We thank the reviewer for his/her very positive comment.

I just have a few questions and minor suggestions:

1. It is not clear how many nanoparticles are being measured. Please give an estimate. How many Eu³⁺ ions in total does this correspond to?

Taking into account the size of the particles, there are about 10^{10} - 10^{11} particles in the sample holder and $5 \cdot 10^{11}$ - $5 \cdot 10^{12}$ Eu³⁺ ions resonant with the laser. However, it is quite difficult to determine what fraction contributes to the signal because of the strong scattering in the powder.

The number of particles has been added to the methods section (p. 9).

2. It would also be useful to quote the average number of Eu³⁺ ions per particle, and the number of magnetic defects per particle inferred from the measurements.

The total number of Eu ions per particle in both crystallographic sites is $4.4 \cdot 10^6$, while the number of inferred defects is estimated to be 200 times lower, $2.2 \cdot 10^4$.

These numbers have been added to the SI on page 8.

3. Have the authors tried to use other dynamical decoupling sequences, such as XY8? Since some sequences are more robust against pulse errors than CPMG, further increase in coherence times could be achieved.

The all-optical DD scheme we used could certainly implement more complex DD sequences than CPMG. We tested XY4 for example but did not find to longer T_2 , possibly because the pulse duration was too long compared to the dephasing process correlation time. As mentioned in our answers to reviewer 1, we feel that further studies along this line would be much more conclusive in a better defined system, like a single particle, in which the spread in Rabi frequencies is smaller.

REVIEWERS' COMMENTS:

Reviewer #1 (Remarks to the Author):

The authors have addressed my questions and I recommend for publication.